# Strictly Batch Imitation Learning
# by Energy-based Distribution Matching

**Daniel Jarrett**[*]
University of Cambridge
daniel.jarrett@maths.cam.ac.uk

**Ioana Bica**[*]
University of Oxford
The Alan Turing Institute
ioana.bica@eng.ox.ac.uk

**Mihaela van der Schaar**
University of Cambridge
University of California, Los Angeles
The Alan Turing Institute
mv472@cam.ac.uk

## Abstract

Consider learning a policy purely on the basis of demonstrated behavior—that is, with no access to reinforcement signals, no knowledge of transition dynamics, and no further interaction with the environment. This *strictly batch imitation learning* problem arises wherever live experimentation is costly, such as in healthcare. One solution is simply to retrofit existing algorithms for apprenticeship learning to work in the offline setting. But such an approach leans heavily on off-policy evaluation or offline model estimation, and can be indirect and inefficient. We argue that a good solution should be able to explicitly parameterize a policy (i.e. respecting action conditionals), implicitly learn from rollout dynamics (i.e. leveraging state marginals), and—crucially—operate in an entirely offline fashion. To address this challenge, we propose a novel technique by *energy-based distribution matching* (EDM): By identifying parameterizations of the (discriminative) model of a policy with the (generative) energy function for state distributions, EDM yields a simple but effective solution that equivalently minimizes a divergence between the occupancy measure for the demonstrator and a model thereof for the imitator. Through experiments with application to control and healthcare settings, we illustrate consistent performance gains over existing algorithms for strictly batch imitation learning.

## 1   Introduction

Imitation learning deals with training an agent to mimic the actions of a demonstrator. In this paper, we are interested in the specific setting of *strictly batch imitation learning*—that is, of learning a policy purely on the basis of demonstrated behavior, with no access to reinforcement signals, no knowledge of transition dynamics, and—importantly—no further interaction with the environment. This problem arises wherever live experimentation is costly, such as in medicine, healthcare, and industrial processes. While behavioral cloning is indeed an intrinsically offline solution as such, it fails to exploit precious information contained in the distribution of states visited by the demonstrator.

Of course, given the rich body of recent work on (online) apprenticeship learning, one solution is simply to repurpose such existing algorithms—including classic inverse reinforcement learning and more recent adversarial imitation learning methods—to operate in the offline setting. However, this strategy leans heavily on off-policy evaluation (which is its own challenge per se) or offline model estimation (inadvisable beyond small or discrete models), and can be indirect and inefficient—via off-policy alternating optimizations, or by running RL in a costly inner loop. Instead, we argue that a good solution should directly parameterize a policy (i.e. respect action conditionals), account for rollout dynamics (i.e. respect state marginals), and—crucially—operate in an entirely offline fashion without recourse to off-policy evaluation for retrofitting existing (but intrinsically online) methods.

**Contributions**   In the sequel, we first formalize imitation learning in the *strictly batch* setting, and motivate the unique desiderata expected of a good solution (Section 2). To meet this challenge, we propose a novel technique by *energy-based distribution matching* (EDM) that identifies parameterizations

---

[*]Authors contributed equally

of the (discriminative) model of a policy with the (generative) energy function for state distributions (Section 3). To understand its relative simplicity and effectiveness for batch learning, we relate the EDM objective to existing notions of divergence minimization, multitask learning, and classical imitation learning (Section 4). Lastly, through experiments with application to control tasks and healthcare, we illustrate consistent improvement over existing algorithms for offline imitation (Section 5).

## 2  Strictly Batch Imitation Learning

**Preliminaries**  We work in the standard Markov decision process (MDP) setting, with states $s \in \mathcal{S}$, actions $a \in \mathcal{A}$, transitions $T \in \Delta(\mathcal{S})^{\mathcal{S} \times \mathcal{A}}$, rewards $R \in \mathbb{R}^{\mathcal{S} \times \mathcal{A}}$, and discount $\gamma$. Let $\pi \in \Delta(\mathcal{A})^{\mathcal{S}}$ denote a policy, with induced occupancy measure $\rho_\pi(s, a) \doteq \mathbb{E}_\pi[\sum_{t=0}^\infty \gamma^t \mathbb{1}_{\{s_t = s, a_t = a\}}]$, where the expectation is understood to be taken over $a_t \sim \pi(\cdot|s_t)$ and $s_{t+1} \sim T(\cdot|s_t, a_t)$ from some initial distribution. We shall also write $\rho_\pi(s) \doteq \sum_a \rho_\pi(s, a)$ to indicate the state-only occupancy measure. In this paper, we operate in the most minimal setting where neither the environment dynamics nor the reward function is known. Classically, *imitation learning* [1–3] seeks an imitator policy $\pi$ as follows:

$$\text{argmin}_\pi \, \mathbb{E}_{s \sim \rho_\pi} \mathcal{L}\big(\pi_D(\cdot|s), \pi(\cdot|s)\big) \tag{1}$$

where $\mathcal{L}$ is some choice of loss. In practice, instead of $\pi_D$ we are given access to a sampled dataset $\mathcal{D}$ of state-action pairs $s, a \sim \rho_D$. (While here we only assume access to *pairs*, some algorithms require *triples* that include next states). Behavioral cloning [4–6] is a well-known (but naive) approach that simply ignores the endogeneity of the rollout distribution, replacing $\rho_\pi$ with $\rho_D$ in the expectation. This reduces imitation learning to a supervised classification problem (popularly, with cross-entropy loss), though the potential disadvantage of disregarding the visitation distribution is well-studied [7–9].

**Apprenticeship Learning**  To incorporate awareness of dynamics, a family of techniques (commonly referenced under the "apprenticeship learning" umbrella) have been developed, including classic inverse reinforcement learning algorithms and more recent methods in adversarial imitation learning. Note that the vast majority of these approaches are *online* in nature, though it is helpful for us to start with the same formalism. Consider the (maximum entropy) reinforcement learning setting, and let $R_t \doteq R(s_t, a_t)$ and $\mathcal{H}_t \doteq -\log \pi(\cdot|s_t)$. The (forward) primitive $\text{RL} : \mathbb{R}^{\mathcal{S} \times \mathcal{A}} \to \Delta(\mathcal{A})^{\mathcal{S}}$ is given by:

$$\text{RL}(R) \doteq \text{argmax}_\pi \left( \mathbb{E}_\pi[\sum_{t=0}^\infty \gamma^t R_t] + H(\pi) \right) \tag{2}$$

where (as before) the expectation is understood to be taken with respect to $\pi$ and the environment dynamics, and $H(\pi) \doteq \mathbb{E}_\pi[\sum_{t=0}^\infty \gamma^t \mathcal{H}_t]$. A basic result [10, 11] is that the (soft) Bellman operator is contractive, so its fixed point (hence the optimal policy) is unique. Now, let $\psi : \mathbb{R}^{\mathcal{S} \times \mathcal{A}} \to \mathbb{R}$ denote a reward function regularizer. Then the (inverse) primitive $\text{IRL}_\psi : \Delta(\mathcal{A})^{\mathcal{S}} \to \mathcal{P}(\mathbb{R}^{\mathcal{S} \times \mathcal{A}})$ is given by:

$$\text{IRL}_\psi(\pi_D) \doteq \text{argmin}_R \left( \psi(R) + \max_\pi \left( \mathbb{E}_\pi[\sum_{t=0}^\infty \gamma^t R_t] + H(\pi) \right) - \mathbb{E}_{\pi_D}[\sum_{t=0}^\infty \gamma^t R_t] \right) \tag{3}$$

Finally, let $\tilde{R} \in \text{IRL}_\psi(\pi_D)$ and $\pi = \text{RL}(\tilde{R})$, and denote by $\psi^* : \mathbb{R}^{\mathcal{S} \times \mathcal{A}} \to \mathbb{R}$ the Fenchel conjugate of regularizer $\psi$. A fundamental result [12] is that ($\psi$-regularized) apprenticeship learning can be taken as the composition of forward and inverse procedures, and obtains an imitator policy $\pi$ such that the induced occupancy measure $\rho_\pi$ is close to $\rho_D$ as determined by the (convex) function $\psi^*$:

$$\text{RL} \circ \text{IRL}_\psi(\pi_D) = \text{argmax}_\pi \left( -\psi^*(\rho_\pi - \rho_D) + H(\pi) \right) \tag{4}$$

Classically, IRL-based apprenticeship methods [13–21] simply execute RL repeatedly in an inner loop, with fixed regularizers $\psi$ for tractability (such as indicators for linear and convex function classes). More recently, adversarial imitation learning techniques leverage Equation 4 (modulo $H(\pi)$, which is generally less important in practice), instantiating $\psi^*$ with various $\phi$-divergences [12, 22–27] and integral probability metrics [28, 29], thereby matching occupancy measures without unnecessary bias.

**Strictly Batch Imitation Learning**  Unfortunately, advances in both IRL-based and adversarial IL have a been developed with a very much *online* audience in mind: Precisely, their execution involves repeated on-policy rollouts, which requires access to an environment (for interaction), or at least knowledge of its dynamics (for simulation). Imitation learning in a completely *offline* setting provides neither. On the other hand, while behavioral cloning is "offline" to begin with, it is fundamentally limited by disregarding valuable (distributional) information in the demonstration data. Proposed

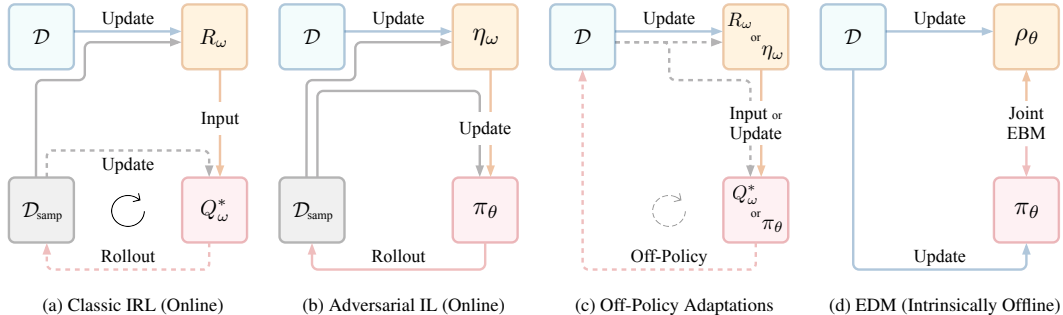

| (a) Classic IRL (Online) | (b) Adversarial IL (Online) | (c) Off-Policy Adaptations | (d) EDM (Intrinsically Offline) |

Figure 1: *From Online to Offline Learning*. **(a)** Classic IRL-based algorithms execute RL repeatedly in an inner loop, learning imitator policies indirectly via parameterizations $\omega$ of a reward function $R_\omega$. **(b)** Adversarial IL methods seek a distribution-matching objective, alternately optimizing a policy $\pi_\theta$ parameterized by $\theta$ and a discriminator-like function $\eta_\omega$ (which in some cases can be taken as $R$ or a value-function) parameterized by $\omega$. **(c)** For strictly batch IL, one solution is simply to retrofit existing algorithms from (a) or (b) to work without any sampling actually taking place; this involves using off-policy evaluation as a workaround for these (intrinsically online) apprenticeship methods, which may introduce more variance than desired. **(d)** We propose a simpler but effective offline method by jointly learning a policy function with an energy-based model of the state distribution.

rectifications are infeasible, as they typically require querying the demonstrator, interacting with the environment, or knowledge of model dynamics or sparsity of rewards [30–33]. Now of course, an immediate question is whether existing apprenticeship methods can be more-or-less repurposed for batch learning (see Figure 1). The answer is certainly yes—but they might not be the most satisfying:

*Adapting Classic IRL*. Briefly, this would inherit the theoretical and computational disadvantages of classic IRL, plus additional difficulties from adapting to batch settings. First, IRL learns imitator policies slowly and indirectly via intermediate parameterizations of $R$, relying on repeated calls to a (possibly imperfect) inner RL procedure. Explicit constraints for tractability also mean that true rewards will likely be imperfectly captured without excessive feature engineering. Most importantly, batch IRL requires *off-policy* evaluation at every step—which is itself a nontrivial problem with imperfect solutions. For instance, for the max-margin, minimax, and max-likelihood approaches, adaptations for batch imitation [34–37] rely on least-squares TD and Q-learning, as well as depending on restrictions to linear rewards. Similarly, adaptations of policy-loss and Bayesian IRL in [34, 38] fall back on linear score-based classification and LSTD. Alternative workarounds involve estimating a model from demonstrations alone [39, 40]—feasible only for the smallest or discrete state spaces.

*Adapting Adversarial IL*. Analogously, the difficulty here is that the adversarial formulation requires expectations over trajectories sampled from imitator policy rollouts. Now, there has been recent work focusing on enabling *off-policy* learning through the use of off-policy actor-critic methods [41, 42]. However, this is accomplished by skewing the divergence minimization objective to minimize the distance between the distributions induced by the demonstrator and the replay buffer (instead of the imitator); they must still operate in an online fashion, and are not applicable in a strictly batch setting. More recently, a reformulation in [43] does away with a separate critic by learning the (log density ratio) "$Q$-function" via the same objective used for distribution matching. While this theoretically enables fully offline learning, it inherits a similarly complex alternating max-min optimization procedure; moreover, the objective involves the logarithm of an expectation over an exponentiated difference in the Bellman operator—for which mini-batch approximations of gradients are biased.

**Three Desiderata** At risk of belaboring, the offline setting means that we already have *all* of the information we will ever get, right at the very start. Hanging on to the RL-centric structure of these intrinsically online apprenticeship methods relies entirely on off-policy techniques—which may introduce more variance than we can afford. In light of the preceding discussion, it is clear that a good solution to the strictly batch imitation learning (SBIL) problem should satisfy the following criteria:

1. **Policy**: First, it should directly learn a policy (capturing "stepwise" action conditionals) without relying on learning intermediate rewards, and without generic constraints biasing the solution.
2. **Occupancy**: But unlike the (purely discriminative) nature of behavioral cloning, it should (generatively) account for information from rollout distributions (capturing "global" state marginals).
3. **Intrinsically Batch**: Finally, it should work offline without known/learned models, and without resorting to off-policy evaluations done within inner loops/max-min optimizations (see Table 1).

# 3 Energy-based Distribution Matching

We begin by parameterizing with $\theta$ our policy $\pi_\theta$, and occupancy measure $\rho_\theta$. We are interested in (explicitly) learning a policy while (implicitly) minimizing a divergence between occupancy measures:

$$\text{argmin}_\theta \, D_\phi(\rho_D \| \rho_\theta) \qquad (5)$$

for some choice of generator $\phi$. Note that, unlike in the case of online apprenticeship, our options are significantly constrained by the fact that rollouts of $\pi_\theta$ are not actually possible. In the sequel, we shall use $\phi(u) = u \log u$, which gives rise to the (forward) KL, so we write $\text{argmin}_\theta \, D_{\text{KL}}(\rho_D \| \rho_\theta) = \text{argmin}_\theta - \mathbb{E}_{s,a \sim \rho_D} \log \rho_\theta(s, a)$. Now, consider the general class of stationary policies of the form:

$$\pi_\theta(a|s) = \frac{e^{f_\theta(s)[a]}}{\sum_a e^{f_\theta(s)[a]}} \qquad (6)$$

where $f_\theta : \mathcal{S} \to \mathbb{R}^{\mathcal{A}}$ indicates the logits for action conditionals. An elementary result [44, 45] shows a bijective mapping between the space of policies and occupancy measures satisfying the Bellman flow constraints, and $\pi(a|s) = \rho_\pi(s, a)/\rho_\pi(s)$; this allows decomposing the log term in the divergence as:

$$\log \rho_\theta(s, a) = \log \rho_\theta(s) + \log \pi_\theta(a|s) \qquad (7)$$

**Objective** Ideally, our desired loss is therefore:

$$\mathcal{L}(\theta) = -\mathbb{E}_{s \sim \rho_D} \log \rho_\theta(s) - \mathbb{E}_{s,a \sim \rho_D} \log \pi_\theta(a|s) \qquad (8)$$

with the corresponding gradient given by:

$$\nabla_\theta \mathcal{L}(\theta) = -\mathbb{E}_{s \sim \rho_D} \nabla_\theta \log \rho_\theta(s) - \mathbb{E}_{s,a \sim \rho_D} \nabla_\theta \log \pi_\theta(a|s) \qquad (9)$$

Now, there is an obvious problem. Backpropagating through the first term is impossible as we cannot compute $\rho_\theta(s)$—nor do we have access to online rollouts of $\pi_\theta$ to explicitly estimate it. In this offline imitation setting, our goal is to answer the question: Is there any benefit in *learning* an approximate model in its place instead? Here we consider energy-based modeling [46], which associates scalar measures of compatibility (i.e. energies) with configurations of variables (i.e. states). Specifically, we take advantage of the *joint energy-based modeling* approach [47–49]—in particular the proposal for a classifier to be simultaneously learned with a density model defined implicitly by the logits of the classifier (which—as they observe—yields improvements such as in calibration and robustness):

**Joint Energy-based Modeling** Consider first the general class of energy-based models (EBMs) for state occupancy measures $\rho_\theta(s) \propto e^{-E(s)}$. Now, mirroring the exposition in [47], note that a model of the state-action occupancy measure $\rho_\theta(s, a) = e^{f_\theta(s)[a]}/Z_\theta$ can be defined via the parameterization for $\pi_\theta$, where $Z_\theta$ is the partition function. The state-only model for $\rho_\theta(s) = \sum_a e^{f_\theta(s)[a]}/Z_\theta$ is then obtained by marginalizing out $a$. In other words, the parameterization of $\pi_\theta$ already implicitly defines an EBM of state visitation distributions with the energy function $E_\theta : \mathbb{R}^{|\mathcal{S}|} \to \mathbb{R}^{|\mathcal{A}|}$ given as follows:

$$E_\theta(s) \doteq -\log \sum_a e^{f_\theta(s)[a]} \qquad (10)$$

The chief difference from [47], of course, is that here the true probabilities in question are not static class conditionals/marginals: The *actual* occupancy measure corresponds to rolling out $\pi_\theta$, and if we could do that, we would naturally recover an approach not unlike the variety of distribution-aware algorithms in the literature; see e.g. [50]. In the strictly batch setting, we clearly cannot sample directly from this (online) distribution. However, as a matter of multitask learning, we still hope to gain from jointly learning an (offline) *model* of the state distribution—which we can then freely sample from:

**Proposition 1 (Surrogate Objective)** Define the "occupancy" loss $\mathcal{L}_\rho$ as the difference in energy:

$$\mathcal{L}_\rho(\theta) \doteq \mathbb{E}_{s \sim \rho_D} E_\theta(s) - \mathbb{E}_{s \sim \rho_\theta} E_\theta(s) \qquad (11)$$

Then $\nabla_\theta \mathcal{L}_\rho(\theta) = -\mathbb{E}_{s \sim \rho_D} \nabla_\theta \log \rho_\theta(s)$. In other words, differentiating this recovers the first term in Equation 9. Therefore if we define a standard "policy" loss $\mathcal{L}_\pi(\theta) \doteq -\mathbb{E}_{s,a \sim \rho_D} \log \pi_\theta(a|s)$, then:

$$\mathcal{L}_{\text{surr}}(\theta) \doteq \mathcal{L}_\rho(\theta) + \mathcal{L}_\pi(\theta) \qquad (12)$$

yields a surrogate objective that can be optimized, instead of the original $\mathcal{L}$. Note that by relying on the offline energy-based model, we now have access to the gradients of the terms in the expectations.

---

**Algorithm 1** Energy-based Distribution Matching       ▷ for Strictly Batch Imitation Learning

1: **Input**: SGLD hyperparameters $\alpha, \sigma$, PCD hyperparameters $\kappa, \iota, \delta$, and mini-batch size $N$
2: **Initialize**: Policy network parameters $\theta$, and PCD buffer $B_\kappa$
3: **while** not converged **do**
4:     Sample $(s_1, a_1), ..., (s_N, a_N) \sim \mathcal{D}$ from demonstrations dataset
5:     Sample $(\tilde{s}_{1,0}, ..., \tilde{s}_{N,0})$ as $\tilde{s}_{n,0} \sim B_\kappa$ **w.p.** $1 - \delta$ **o.w.** $\tilde{s}_{n,0} \sim \mathcal{U}(\mathcal{S})$
6:     **for** $i = 1, ..., \iota$ **do**
7:         $\tilde{s}_{n,i} = \tilde{s}_{n,i-1} - \alpha \cdot \partial E_\theta(\tilde{s}_{n,i-1})/\partial \tilde{s}_{n,i-1} + \sigma \cdot \mathcal{N}(0, I), \forall n \in \{1, ..., N\}$
8:     $\hat{\mathcal{L}}_\pi \leftarrow \frac{1}{N} \sum_{n=1}^{N} \text{CrossEntropy}(\pi_\theta(\cdot|s_n), a_n)$         ▷ $\mathcal{L}_\pi = -\mathbb{E}_{s,a \sim \rho_D} \log \pi_\theta(a|s)$
9:     $\hat{\mathcal{L}}_\rho \leftarrow \frac{1}{N} \sum_{n=1}^{N} E_\theta(s_n) - \frac{1}{N} \sum_{n=1}^{N} E_\theta(\tilde{s}_{n,\iota})$     ▷ $\mathcal{L}_\rho = \mathbb{E}_{s \sim \rho_D} E_\theta(s) - \mathbb{E}_{s \sim \rho_\theta} E_\theta(s)$
10:     Add $\tilde{s}_{n,\iota}$ to $B_\kappa, \forall n \in \{1, ..., N\}$
11:     Backpropagate $\nabla_\theta \hat{\mathcal{L}}_\rho + \nabla_\theta \hat{\mathcal{L}}_\pi$
12: **Output**: Learned policy parameters $\theta$

---

*Proof.* Appendix A. Sketch: For any $s$, write $\rho_\theta(s) = e^{-E_\theta(s)} / \int_{\mathcal{S}} e^{-E_\theta(s)} ds$, for which the gradient of the logarithm is given by $-\nabla_\theta \log \rho_\theta(s) = \nabla_\theta E_\theta(s) - \mathbb{E}_{s \sim \rho_\theta} \nabla_\theta E_\theta(s)$. Then, taking expectations over $\rho_D$ and substituting in the energy term as given by Equation 10, straightforward manipulation shows $-\nabla_\theta \mathbb{E}_{s \sim \rho_D} \log \rho_\theta(s) = \nabla_\theta \mathcal{L}_\rho(\theta)$. The second part follows immediately from Equation 8. $\square$

Why is this better than before? Because using the original objective $\mathcal{L}$ required us to know $\rho_\theta(s)$, which—even modeled separately as an EBM—we do not (since we cannot compute the normalizing constant). On the other hand, using the surrogate objective $\mathcal{L}_{\text{surr}}$ only requires being able to sample from the EBM, which is easier. Note that jointly learning the EBM does not constrain/bias the policy, as this simply reuses the policy parameters along with the extra degree of freedom in the logits $f_\theta(s)[\cdot]$.

**Optimization** The EDM surrogate objective entails minimal addition to the standard behavioral cloning loss. Accordingly, it is perfectly amenable to mini-batch gradient approximations—unlike for instance [43], for which mini-batch gradients are biased in general. We approximate the expectation over $\rho_\theta$ in Equation 11 via stochastic gradient Langevin dynamics (SGLD) [51], which follows recent successes in training EBMs parameterized by deep networks [47, 48, 52], and use persistent contrastive divergence (PCD) [53] for computational savings. Specifically, each sample is drawn as:

$$\tilde{s}_i = \tilde{s}_{i-1} - \alpha \cdot \frac{\partial E_\theta(\tilde{s}_{i-1})}{\partial \tilde{s}_{i-1}} + \sigma \cdot \mathcal{N}(0, I) \tag{13}$$

where $\alpha$ denotes the SGLD learning rate, and $\sigma$ the noise coefficient. Algorithm 1 details the EDM optimization procedure, with a buffer $B_\kappa$ of size $\kappa$, reinitialization frequency $\delta$, and number of iterations $\iota$, where $\tilde{s}_0 \sim \rho_0(s)$ is sampled uniformly. Note that the buffer here should not be confused with the "replay buffer" within (online) imitation learning algorithms, to which it bears no relationship whatsoever. In practice, we find that the configuration given in [47] works effectively with only small modifications. We refer to [46–49, 51, 53] for discussion of general considerations for EBM optimization.

## 4 Analysis and Interpretation

Our development in Section 3 proceeded in three steps. First, we set out with a divergence minimization objective in mind (Equation 5). With the aid of the decomposition in Equation 7, we obtained the original (online) maximum-likelihood objective function (Equation 8). Finally, using Proposition 1, we instead optimize an (offline) joint energy-based model by scaling the gradient of a surrogate objective (Equation 12). Now, the mechanics of the optimization are straightforward, but what is the underlying motivation for doing so? In particular, how does the EDM objective relate to existing notions of (1) divergence minimization, (2) joint learning, as well as (3) imitation learning in general?

**Divergence Minimization** With the seminal observation by [12] of the equivalence in Equation 4, the IL arena was quickly populated with a lineup of adversarial algorithms minimizing a variety of distances [25–29], and the forward KL in this framework was first investigated in [27]. However in the strictly batch setting, we have no ability to compute (or even sample from) the actual rollout distribution for $\pi_\theta$, so we instead choose to learn an EBM in its place. To be clear, we are now doing something quite different than [25–29]: In minimizing the divergence (Equation 5) by simultaneously learning an (offline) model instead of sampling from (online) rollouts, $\pi_\theta$ and $\rho_\theta$ are no longer coupled in terms of rollout *dynamics*, and the coupling that remains is in terms of the underlying *parameterization* $\theta$. That is the price we pay. At the same time, hanging on to the adversarial setup in the batch setting requires

Table 1: *From Online to Offline Imitation*. Recall the three desiderata from Section 2, where we seek an SBIL solution that: **(1)** learns a *directly parameterized* policy, without restrictive constraints biasing the solution—e.g. restrictions to linear/convex function classes for intermediate rewards, or generic norm-based penalties on reward sparsity; **(2)** is *dynamics-aware* by accounting for distributional information—either through temporal or parameter consistency; and **(3)** is *intrinsically batch*, in the sense of being operable strictly offline, and directly optimizable—i.e. without recourse to off-policy evaluations in costly inner loops or alternating max-min optimizations.

| | Formulation | Example | Parameterized Policy[1] | Non-Restrictive Regularizer[1] | Dynamics Awareness[2] | Operable Strictly Batch[3] | Directly Optimized[3] |
|---|---|---|---|---|---|---|---|
| Online (Original) | Max Margin | [13,14] | ✗ | ✗ | ✓ | ✗ | ✗ |
| | Minimax Game | [17] | ✗ | ✗ | ✓ | ✗ | ✗ |
| | Min Policy Loss | [15] | ✗ | ✗ | ✓ | ✗ | ✗ |
| | Max Likelihood | [19] | ✗ | ✗ | ✓ | ✗ | ✗ |
| | Max Entropy | [10,18] | ✗ | ✗ | ✓ | ✗ | ✗ |
| | Max A Posteriori | [16,20] | ✗ | ✗ | ✓ | ✗ | ✗ |
| | Adversarial Imitation | [12,22-27] | ✓ | ✓ | ✓ | ✗ | ✗ |
| Off. (Adaptation) | Max Margin | [34,37] | ✗ | ✗ | ✓ | ✓ | ✗ |
| | Minimax Game | [35] | ✗ | ✗ | ✓ | ✓ | ✗ |
| | Min Policy Loss | [54] | ✗ | ✗ | ✓ | ✓ | ✓ |
| | Max Likelihood | [36] | ✗ | ✗ | ✓ | ✓ | ✗ |
| | Max Entropy | [39] | ✗ | ✗ | ✓ | ✓ | ✗ |
| | Max A Posteriori | [38] | ✗ | ✗ | ✓ | ✓ | ✓ |
| | Adversarial Imitation | [43] | ✓ | ✓ | ✓ | ✓ | ✗ |
| | Behavioral Cloning | [7] | ✓ | ✗ | ✗ | ✓ | ✓ |
| | Reward-Regularized BC | [9] | ✓ | ✗ | ✓ | ✓ | ✓ |
| | **EDM** | (Ours) | ✓ | ✓ | ✓ | ✓ | ✓ |

estimating intrinsically on-policy terms via off-policy methods, which are prone to suffer from high variance. Moreover, the divergence minimization interpretations of adversarial IL hinge crucially on the assumption that the discriminator-like function is perfectly optimized [12,25,27,43]—which may not be realized in practice offline. The EDM objective aims to sidestep both of these difficulties.

**Joint Learning** In the online setting, minimizing Equation 8 is equivalent to injecting temporal consistency into behavioral cloning: While the $\mathbb{E}_{s,a\sim\rho_D}\log\pi_\theta(a|s)$ term is purely a discriminative objective, the $\mathbb{E}_{s\sim\rho_D}\log\rho_\theta(s)$ term additionally constrains $\pi_\theta(\cdot|s)$ to the space of policies for which the induced state distribution matches the data. In the offline setting, instead of this *temporal* relationship we are now leveraging the *parameter* relationship between $\pi_\theta$ and $\rho_\theta$—that is, from the joint EBM. In effect, this accomplishes an objective similar to multitask learning, where representations of both discriminative (policy) and generative (visitation) distributions are learned by sharing the same underlying function approximator. As such, (details of sampling techniques aside) this additional mandate does *not* add any bias. This is in contrast to generic approaches to regularization in IL, such as the norm-based penalties on the sparsity of implied rewards [9,32,55]—which adds bias. The state-occupancy constraint in EDM simply harnesses the extra degree of freedom hidden in the logits $f_\theta(s)$—which are normally allowed to shift by an arbitrary scalar—to define the density over states.

**Imitation Learning** Finally, recall the classical notion of *imitation learning* that we started with (Equation 1). As noted earlier, naive application by behavioral cloning simply ignores the endogeneity of the rollout distribution. How does our final surrogate objective (Equation 12) relate to this? First, we place Equation 1 in the maximum entropy RL framework in order to speak in a unified language:

**Proposition 2 (Classical Objective)** Consider the classical IL objective in Equation 1, with policies parameterized as Equation 6. Choosing $\mathcal{L}$ to be the (forward) KL divergence yields the following:

$$\mathrm{argmax}_R \left( \mathbb{E}_{s\sim\rho_R^*}\mathbb{E}_{a\sim\pi_D(\cdot|s)}Q_R^*(s,a) - \mathbb{E}_{s\sim\rho_R^*}V_R^*(s) \right) \qquad (14)$$

where $Q_R^* : \mathcal{S} \times \mathcal{A} \to \mathbb{R}$ is the (soft) $Q$-function given by $Q_R^*(s,a) = R(s,a) + \gamma\mathbb{E}_T[V_R^*(s')|s,a]$, $V^*(s) \in \mathbb{R}^{\mathcal{S}}$ is the (soft) value function $V_R^*(s) = \log\sum_a e^{Q_R^*(s,a)}$, and $\rho_R^*$ is the occupancy for $\pi_R^*$.

*Proof.* Appendix A. This relies on the fact that we are free to identify the logits $f_\theta$ of our policy with a (soft) $Q$-function. Specifically, this requires the additional fact that the mapping between $Q$-functions and reward functions is bijective, which we also state (and prove) as Lemma 5 in Appendix A. □

This is intuitive: It states that classical imitation learning with $\mathcal{L} = D_{\mathrm{KL}}$ is equivalent to searching for a reward function $R$. In particular, we are looking for an $R$ for which—in expectation over *rollouts* of policy $\pi_R^*$—the advantage function $Q_R^*(s,a) - V_R^*(s)$ for taking actions $a \sim \pi_D(\cdot|s)$ is

maximal. Now, the following distinction is key: While Equation 14 is perfectly valid as a choice of objective, it is a certain (naive) substitution in the offline setting that is undesirable. Specifically, Equation 14 is precisely what behavioral cloning attempts to do, but—without the ability to perform $\pi_R^*$ rollouts—it simply replaces $\rho_R^*$ with $\rho_D$. This is *not* an (unbiased) "approximation" in, say, the sense that $\hat{\mathcal{L}}_\rho$ empirically approximates $\mathcal{L}_\rho$, and is especially inappropriate when $\rho_D$ contains very few demonstrations to begin with. While EDM cannot fully "undo" the damage (nothing can do that in the strictly batch setting), it uses a "smoothed" EBM in place of $\rho_D$, which—as we shall see empirically—leads to largest improvements precisely when the number of demonstrations are few.

**Proposition 3 (From BC to EDM)** The behavioral cloning objective is equivalently the following, where—compared to Equation 14—expectations over states are now taken w.r.t. $\rho_D$ instead of $\rho_R^*$:

$$\operatorname{argmax}_R \left( \mathbb{E}_{s\sim\rho_D} \mathbb{E}_{a\sim\pi_D(\cdot|s)} Q_R^*(s,a) - \mathbb{E}_{s\sim\rho_D} V_R^*(s) \right) \tag{15}$$

In contrast, by augmenting the (behavioral cloning) "policy" loss $\mathcal{L}_\pi$ with the "occupancy" loss $\mathcal{L}_\rho$, what the EDM surrogate objective achieves is to replace one of the expectations with the learned $\rho_\theta$:

$$\operatorname{argmax}_R \left( \mathbb{E}_{s\sim\rho_D} \mathbb{E}_{a\sim\pi_D(\cdot|s)} Q_R^*(s,a) - \mathbb{E}_{s\sim\rho_\theta} V_R^*(s) \right) \tag{16}$$

*Proof.* Appendix A. The reasoning for both statements follows a similar form as for Proposition 2. $\square$

Note that by swapping out $\rho_R^*$ for $\rho_D$ in behavioral cloning, the (dynamics) relationship between $\pi_R^*$ and its induced occupancy measure is (completely) broken, and the optimization in Equation 15 is equivalent to performing a sort of inverse reinforcement learning with no constraints whatsoever on $R$. What the EDM surrogate objective does is to "repair" one of the expectations to allow sampling from a smoother model distribution $\rho_\theta$ than the (possibly very sparse) data distribution $\rho_D$. (Can we also "repair" the other term? But this is now asking to somehow warp $\mathbb{E}_{s\sim\rho_D}\mathbb{E}_{a\sim\pi_D(\cdot|s)}Q_R^*(s,a)$ into $\mathbb{E}_{s\sim\rho_\theta}\mathbb{E}_{a\sim\pi_D(\cdot|s)}Q_R^*(s,a)$. All else equal, this is certainly impossible without querying the expert.)

## 5 Experiments

**Benchmarks** We test Algorithm 1 (**EDM**) against the following benchmarks, varying the amount of demonstration data $\mathcal{D}$ (from a single trajectory to 15) to illustrate sample complexity: The intrinsically offline behavioral cloning (**BC**), and reward-regularized classification (**RCAL**) [32]—which proposes to leverage dynamics information through a sparsity-based penalty on the implied rewards; the deep successor feature network (**DFSN**) algorithm of [37]—which is an off-policy adaptation of the max-margin IRL algorithm and a (deep) generalization of earlier (linear) approaches by LSTD [34,38]; and the state-of-the-art in sample-efficient adversarial imitation learning (**VDICE**) in [43], which—while designed with an online audience in mind—can theoretically operate in a completely offline manner. (Remaining candidates in Table 1 are inapplicable, since they either only operate in discrete states [36,39], or only output a reward [54], which—in the strictly batch setting—does not yield a policy.

**Demonstrations** We conduct experiments on control tasks and a real-world healthcare dataset. For the former, we use OpenAI gym environments [56] of varying complexity from standard RL literature: CartPole, which balances a pendulum on a frictionless track [57], Acrobot, which swings a system of joints up to a given height [58], BeamRider, which controls an Atari 2600 arcade space shooter [59], as well as LunarLander, which optimizes a rocket trajectory for successful landing [60]. Demonstration datasets $\mathcal{D}$ are generated using pre-trained and hyperparameter-optimized agents from the RL Baselines Zoo [61] in Stable OpenAI Baselines [62]. For the healthcare application, we use MIMIC-III, a real-world medical dataset consisting of patients treated in intensive care units from the Medical Information Mart for Intensive Care [63], which records trajectories of physiological states and treatment actions (e.g. antibiotics and ventilator support) for patients at one-day intervals.

**Implementation** The experiment is arranged as follows: Demonstrations $\mathcal{D}$ are sampled for use as input to train all algorithms, which are then evaluated using 300 live episodes (for OpenAI gym environments) or using a held-out test set (for MIMIC-III). This process is then repeated for a total 50 times (using different $\mathcal{D}$ and randomly initialized networks), from which we compile the means of the performances (and their standard errors) for each algorithm. Policies trained by all algorithms share the same network architecture: two hidden layers of 64 units each with ELU activation (or—for Atari—three convolutional layers with ReLU activation). For DSFN, we use the publicly available source code at [64], and likewise for VDICE, which is available at [65]. Note that VDICE is originally designed for Gaussian actions, so we replace the output layer of the actor with a Gumbel-softmax parameterization; offline learning is enabled by setting the "replay regularization" coefficient to zero.

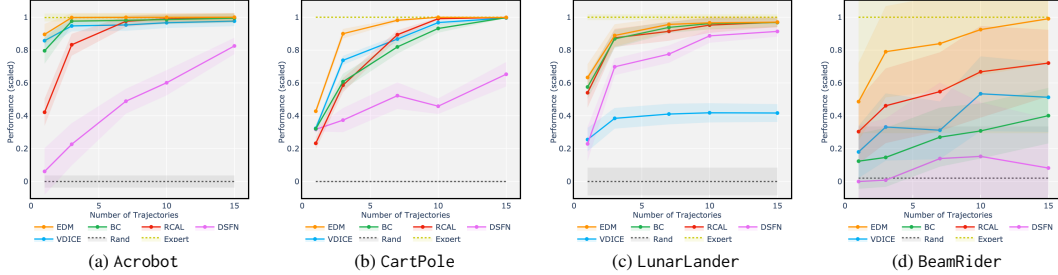

Figure 2: *Performance Comparison for Gym Environments*. The $x$-axis indicates the amount of demonstration data provided (i.e. number of trajectories, in $\{1,3,7,10,15\}$), and the $y$-axis shows the average returns of each imitation algorithm (scaled so that the demonstrator attains a return of 1 and a random policy network attains 0).

| | 2-Action Setting (Ventilator Only) | | | 4-Action Setting (Antibiotics + Vent.) | | |
|---|---|---|---|---|---|---|
| *Metrics* | ACC | AUC | APR | ACC | AUC | APR |
| BC | $0.861 \pm 0.013$ | $0.914 \pm 0.003$ | $0.902 \pm 0.005$ | $0.696 \pm 0.006$ | $0.859 \pm 0.003$ | $0.659 \pm 0.007$ |
| RCAL | $0.872 \pm 0.007$ | $0.911 \pm 0.007$ | $0.898 \pm 0.006$ | $0.701 \pm 0.007$ | $0.864 \pm 0.003$ | $0.667 \pm 0.006$ |
| DSFN | $0.865 \pm 0.007$ | $0.906 \pm 0.003$ | $0.885 \pm 0.001$ | $0.682 \pm 0.005$ | $0.857 \pm 0.002$ | $0.665 \pm 0.003$ |
| VDICE | $0.875 \pm 0.004$ | $0.915 \pm 0.001$ | $0.904 \pm 0.002$ | $0.707 \pm 0.005$ | $0.864 \pm 0.002$ | $0.673 \pm 0.003$ |
| Rand | $0.498 \pm 0.007$ | $0.500 \pm 0.000$ | $0.500 \pm 0.000$ | $0.251 \pm 0.005$ | $0.500 \pm 0.000$ | $0.250 \pm 0.000$ |
| **EDM** | $\mathbf{0.891 \pm 0.004}$ | $\mathbf{0.922 \pm 0.004}$ | $\mathbf{0.912 \pm 0.005}$ | $\mathbf{0.720 \pm 0.007}$ | $\mathbf{0.873 \pm 0.002}$ | $\mathbf{0.681 \pm 0.008}$ |

Table 2: *Performance Comparison for* `MIMIC-III`. Action-matching is used to assess the quality of clinical policies learned in both the 2-action and 4-action settings. We report the accuracy of action selection (ACC), the area under the receiving operator characteristic curve (AUC), and the area under the precision-recall curve (APR).

Algorithm 1 is implemented using the source code for joint EBMs [47] publicly available at [66], which already contains an implementation of SGLD. Note that the only difference between BC and EDM is the addition of $\mathcal{L}_\rho$, and the RCAL loss is straightforwardly obtained by inverting the Bellman equation. See Appendix B for additional detail on experiment setup, benchmarks, and environments.

**Evaluation and Results** For gym environments, the performance of trained imitator policies (learned offline) is evaluated with respect to (true) *average returns* generated by deploying them live. Figure 2 shows the results for policies given different numbers of trajectories as input to training, and Appendix B provides exact numbers. For the `MIMIC-III` dataset, policies are trained and tested on demonstrations by way of cross-validation; since we have no access to ground-truth rewards, we assess performance according to *action-matching* on held-out test trajectories, per standard [64]; Table 2 shows the results. With respect to either metric, we find that EDM consistently produces policies that perform similarly or better than benchmark algorithms in all environments, especially in low-data regimes. Also notable is that in this strictly batch setting (i.e. where no online sampling whatsoever is permitted), the off-policy adaptations of online algorithms (i.e. DSFN, VDICE) do not perform as consistently as the intrinsically offline ones—especially DSFN, which involves predicting entire next states (off-policy) for estimating feature maps; this validates some of our original motivations. Finally, note that—via the joint EBM—the EDM algorithm readily accommodates (semi-supervised) learning from additional state-only data (with unobserved actions); additional result in Appendix B.

## 6 Discussion

In this work, we motivated and presented EDM for strictly batch imitation, which retains the simplicity of direct policy learning while accounting for information in visitation distributions. However, we are sampling from an offline model (leveraging multitask learning) of state visitations, not from actual online rollouts (leveraging temporal consistency), so they can only be so useful. The objective also relies on the assumption that samples in $\mathcal{D}$ are sufficiently representative of $\rho_D$; while this is standard in literature [42], it nonetheless bears reiteration. Our method is agnostic as to discrete/continuous state spaces, but the use of joint EBMs means we only consider categorical actions in this work. That said, the application of EBMs to regression is increasingly of focus [67], and future work may investigate the possibility of extending EDM to continuous actions. Overall, our work is enabled by recent advances in joint EBMs, and similarly use contrastive divergence to approximate the KL gradient. Note that EBMs in general may not be the easiest to train, or to gauge learning progress for [47]. However, for the types of environments we consider, we did not find stability-related issues to be nearly as noticeable as typical of the higher-dimensional imaging tasks EBMs are commonly used for.

## Broader Impact

In general, any method for imitation learning has the potential to mitigate problems pertaining to scarcity of expert knowledge and computational resources. For instance, consider a healthcare institution strapped for time and personnel attention—such as one under the strain of an influx of ICU patients. If implemented as a system for clinical decision support and early warnings, even the most bare-bones policy trained on optimal treatment/monitoring actions has huge potential for streamlining medical decisions, and for allocating attention where real-time clinical judgment is most required.

By focusing our work on the strictly batch setting for learning, we specifically accommodate situations that disallow directly experimenting on the environment during the learning process. This consideration is critical in many conceivable applications: In practice, humans are often on the receiving end of actions and polices, and an imitator policy that must learn by interactive experimentation would be severely hampered due to considerations of cost, danger, or moral hazard. While—in line with literature—we illustrate the technical merits of our proposed method with respect to standard control environments, we do take care to highlight the broader applicability of our approach to healthcare settings, as it likewise applies—without saying—to education, insurance, or even law enforcement.

Of course, an important caveat is that any method for imitation learning naturally runs the risk of internalizing any existing human biases that may be implicit in the demonstrations collected as training input. That said, a growing field in reinforcement learning is dedicated to maximizing interpretability in learned policies, and—in the interest of accountability and transparency—striking an appropriate balance with performance concerns will be an interesting direction of future research.

## Acknowledgments

We would like to thank the reviewers for their generous and invaluable comments and suggestions. This work was supported by Alzheimer's Research UK (ARUK), The Alan Turing Institute (ATI) under the EPSRC grant EP/N510129/1, The US Office of Naval Research (ONR), and the National Science Foundation (NSF) under grant numbers 1407712, 1462245, 1524417, 1533983, and 1722516.

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
