[Supplementary Material]

# A Proofs of Propositions

**Lemma 4** Let $\theta \in \Theta$ be some parameter, consider a random variable $s \in \mathcal{S}$, and fix $f : \mathcal{S} \times \Theta \to \mathbb{R}$, where $f(s, \theta)$ is continuously differentiable with respect to $\theta$ and integrable for all $\theta$. Assume for some random variable $X$ with finite mean that $|\frac{\partial}{\partial \theta} f(s, \theta)| \leq X$ holds almost surely for all $\theta$. Then:

$$\frac{\partial}{\partial \theta} \mathbb{E}[f(s, \theta)] = \mathbb{E}[\frac{\partial}{\partial \theta} f(s, \theta)] \tag{17}$$

*Proof.* $\frac{\partial}{\partial \theta} \mathbb{E}[f(s, \theta)] = \lim_{\delta \to 0} \frac{1}{\delta} (\mathbb{E}[f(s, \theta + \delta)] - \mathbb{E}[f(s, \theta)]) = \lim_{\delta \to 0} \mathbb{E}[\frac{1}{\delta}(f(s, \theta + \delta) - f(s, \theta))]$
$= \lim_{\delta \to 0} \mathbb{E}[\frac{\partial}{\partial \theta} f(s, \tau(\delta))] = \mathbb{E}[\lim_{\delta \to 0} \frac{\partial}{\partial \theta} f(s, \tau(\delta))] = \mathbb{E}[\frac{\partial}{\partial \theta} f(s, \theta)]$, where for the third equality the mean value theorem guarantees the existence of $\tau(\delta) \in (\theta, \theta + \delta)$, and the fourth equality uses the dominated convergence theorem where $|\frac{\partial}{\partial \theta} f(s, \tau(\delta))| \leq X$ by assumption. Note that generalizing to the multivariate case (i.e. gradients) simply requires that the bound be on $\max_i |\frac{\partial}{\partial \theta_i} f(s, \theta)|$ for elements $i$ of $\theta$. Note that most machine learning models (and energy-based models) meet/assume these regularity conditions or similar variants; see e.g. discussion presented in Section 18.1 in [68].

**Proposition 1 (Surrogate Objective)** Define the "occupancy" loss $\mathcal{L}_\rho$ as the difference in energy:

$$\mathcal{L}_\rho(\theta) \doteq \mathbb{E}_{s \sim \rho_D} E_\theta(s) - \mathbb{E}_{s \sim \rho_\theta} E_\theta(s) \tag{11}$$

Then $\nabla_\theta \mathcal{L}_\rho(\theta) = -\mathbb{E}_{s \sim \rho_D} \nabla_\theta \log \rho_\theta(s)$. In other words, differentiating this recovers the first term in Equation 9. Therefore if we define a standard "policy" loss $\mathcal{L}_\pi(\theta) \doteq -\mathbb{E}_{s, a \sim \rho_D} \log \pi_\theta(a|s)$, then:

$$\mathcal{L}_{\text{surr}}(\theta) \doteq \mathcal{L}_\rho(\theta) + \mathcal{L}_\pi(\theta) \tag{12}$$

yields a surrogate objective that can be optimized, instead of the original $\mathcal{L}$. Note that by relying on the offline energy-based model, we now have access to the gradients of the terms in the expectations.

*Proof.* For each $s$, first write the state occupancy measure as $\rho_\theta(s) = e^{-E_\theta(s)} / \int_\mathcal{S} e^{-E_\theta(s)} ds$, so:

$$-\log \rho_\theta(s) = E_\theta(s) + \log \int_\mathcal{S} e^{-E_\theta(s)} ds \tag{18}$$

with gradients given by:

$$-\nabla_\theta \log \rho_\theta(s) = \nabla_\theta E_\theta(s) + \nabla_\theta \log \int_\mathcal{S} e^{-E_\theta(s)} ds$$
$$= \nabla_\theta E_\theta(s) - \frac{\int_\mathcal{S} \nabla_\theta E_\theta(s) e^{-E_\theta(s)} ds}{\int_\mathcal{S} e^{-E_\theta(s)} ds}$$
$$= \nabla_\theta E_\theta(s) - \mathbb{E}_{s \sim \rho_\theta} \nabla_\theta E_\theta(s) \tag{19}$$

Then taking expectations over $\rho_D$ and substituting in the energy term per Equation 10, we have that:

$$-\nabla_\theta \mathbb{E}_{s \sim \rho_D} \log \rho_\theta(s) = \mathbb{E}_{s \sim \rho_D} \big[ \nabla_\theta E_\theta(s) - \mathbb{E}_{s \sim \rho_\theta} \nabla_\theta E_\theta(s) \big]$$
$$= \mathbb{E}_{s \sim \rho_D} \nabla_\theta E_\theta(s) - \mathbb{E}_{s \sim \rho_\theta} \nabla_\theta E_\theta(s)$$
$$= \mathbb{E}_{s \sim \rho_\theta} \nabla_\theta \big( \log \sum_a e^{f_\theta(s)[a]} \big) - \mathbb{E}_{s \sim \rho_D} \nabla_\theta \big( \log \sum_a e^{f_\theta(s)[a]} \big)$$
$$= \nabla_\theta \big( \mathbb{E}_{s \sim \rho_\theta} \log \sum_a e^{f_\theta(s)[a]} - \mathbb{E}_{s \sim \rho_D} \log \sum_a e^{f_\theta(s)[a]} \big)$$
$$= \nabla_\theta \mathcal{L}_\rho(\theta) \tag{20}$$

where the fourth equality uses Lemma 4. Hence we can define $\mathcal{L}_\rho(\theta) \doteq \mathbb{E}_{s \sim \rho_D} E_\theta(s) - \mathbb{E}_{s \sim \rho_\theta} E_\theta(s)$ in lieu of the first term in Equation 8. However, note that the (gradient-based) implementation of Algorithm 1 works even without first obtaining an expression for $\mathcal{L}_\rho(\theta)$ per se, and is correct due to a simpler reason: The batched (empirical loss) $\nabla_\theta \hat{\mathcal{L}}_\rho$ portion of the update (Line 9) is directly analogous to the gradient update in standard contrastive divergence; see e.g. Section 18.2 in [68]. $\square$

Propositions 2–3 first require an additional lemma that allows moving freely between the space of (soft) $Q$-functions and reward functions. Recall the (soft) Bellman operator $\mathbb{B}_R^* : \mathbb{R}^{\mathcal{S} \times \mathcal{A}} \to \mathbb{R}^{\mathcal{S} \times \mathcal{A}}$:

$$(\mathbb{B}_R^* Q)(s, a) = R(s, a) + \gamma \mathbb{E}_T[\text{softmax}_{a'} Q(s', a')|s, a] \tag{21}$$

where $\text{softmax}_a Q(s, a) \doteq \log \sum_a e^{Q(s, a)}$. We know that $\mathbb{B}_R^*$ is contractive with $Q_R^*$ its unique fixed point [10, 11]. Now, let us define the (soft) inverse Bellman operator $\mathbb{J}^* : \mathbb{R}^{\mathcal{S} \times \mathcal{A}} \to \mathbb{R}^{\mathcal{S} \times \mathcal{A}}$ such that:

$$(\mathbb{J}^* Q)(s, a) = Q(s, a) - \gamma \mathbb{E}_T[\text{softmax}_a Q(s', a')|s, a] \tag{22}$$

**Lemma 5** The operator $\mathbb{J}^*$ is *bijective*: $Q = Q_R^* \Leftrightarrow \mathbb{J}^* Q = R$, hence we can write $(\mathbb{J}^*)^{-1} R = Q_R^*$. This is the "soft" version of an analogous statement made for "hard" optimality first shown in [32].

*Proof.* By the uniqueness of the fixed point of $\mathbb{B}_R^*$, we have that $R = \mathbb{J}^* Q \Leftrightarrow \mathbb{B}_R^* Q = Q \Leftrightarrow Q = Q_R^*$. Therefore the inverse image of every singleton $R \in \mathbb{R}^{\mathcal{S} \times \mathcal{A}}$ must exist, and is uniquely equal to $Q_R^*$. This argument is the direct counterpart to Theorem 2 in [32]—which uses argmax instead of softmax.

**Proposition 2 (Classical Objective)** Consider the classical IL objective in Equation 1, with policies parameterized as Equation 6. Choosing $\mathcal{L}$ to be the (forward) KL divergence yields the following:

$$\text{argmax}_R \left( \mathbb{E}_{s \sim \rho_R^*} \mathbb{E}_{a \sim \pi_D(\cdot|s)} Q_R^*(s,a) - \mathbb{E}_{s \sim \rho_R^*} V_R^*(s) \right) \tag{14}$$

where $Q_R^* : \mathcal{S} \times \mathcal{A} \to \mathbb{R}$ is the (soft) Q-function given by $Q_R^*(s,a) = R(s,a) + \gamma \mathbb{E}_T[V_R^*(s')|s,a]$, $V^*(s) \in \mathbb{R}^{\mathcal{S}}$ is the (soft) value function $V_R^*(s) = \log \sum_a e^{Q_R^*(s,a)}$, and $\rho_R^*$ is the occupancy for $\pi_R^*$.

*Proof.* From Equations 1 and 6, choosing $\mathcal{L}$ to be the forward KL divergence yields the following:

$$\text{argmax}_\theta \left( \mathbb{E}_{s \sim \rho_\theta} \mathbb{E}_{a \sim \pi_D(\cdot|s)} f_\theta(s)[a] - \mathbb{E}_{s \sim \rho_\theta} \log \sum_a e^{f_\theta(s)[a]} \right) \tag{23}$$

Now, observe that we are free to identify the logits $f_\theta(s)[a] \in \mathbb{R}^{\mathcal{S} \times \mathcal{A}}$ with a (soft) Q-function. Specifically, define $Q(s,a) \doteq f_\theta(s)[a]$ for all $s, a \in \mathcal{S} \times \mathcal{A}$. Then by Lemma 5 we know there exists a unique $R \in \mathbb{R}^{\mathcal{S} \times \mathcal{A}}$ that $\mathbb{J}^*$ takes $Q$ to. Hence $f_\theta(s)[a] = Q_R^*(s,a)$ for some $R$, and we can write:

$$\text{argmax}_R \left( \mathbb{E}_{s \sim \rho_R^*} \mathbb{E}_{a \sim \pi_D(\cdot|s)} Q_R^*(s,a) - \mathbb{E}_{s \sim \rho_R^*} \log \sum_a e^{Q_R^*(s,a)} \right) \tag{24}$$

where $\pi_R^*(a|s) = e^{Q_R^*(s,a) - V_R^*(s)}$. Then Proposition 2 follows, since $V_R^*(s) = \log \sum_a e^{Q_R^*(s,a)}$. $\square$

**Proposition 3 (From BC to EDM)** The behavioral cloning objective is equivalently the following, where—compared to Equation 14—expectations over states are now taken w.r.t. $\rho_D$ instead of $\rho_R^*$:

$$\text{argmax}_R \left( \mathbb{E}_{s \sim \rho_D} \mathbb{E}_{a \sim \pi_D(\cdot|s)} Q_R^*(s,a) - \mathbb{E}_{s \sim \rho_D} V_R^*(s) \right) \tag{15}$$

In contrast, by augmenting the (behavioral cloning) "policy" loss $\mathcal{L}_\pi$ with the "occupancy" loss $\mathcal{L}_\rho$, what the EDM surrogate objective achieves is to replace one of the expectations with the learned $\rho_\theta$:

$$\text{argmax}_R \left( \mathbb{E}_{s \sim \rho_D} \mathbb{E}_{a \sim \pi_D(\cdot|s)} Q_R^*(s,a) - \mathbb{E}_{s \sim \rho_\theta} V_R^*(s) \right) \tag{16}$$

*Proof.* By definition of behavioral cloning, the only difference is that the expectation in Equation 1 is taken over $\rho_D$; then the same argument for Proposition 2 applies. As for EDM, from Equation 12:

$$\begin{aligned}
\mathcal{L}_{\text{surr}}(\theta) &= \mathcal{L}_\rho(\theta) + \mathcal{L}_\pi(\theta) \\
&= \mathbb{E}_{s \sim \rho_D} E_\theta(s) - \mathbb{E}_{s \sim \rho_\theta} E_\theta(s) - \mathbb{E}_{s,a \sim \rho_D} \log \pi_\theta(a|s) \\
&= \mathbb{E}_{s \sim \rho_\theta} \log \sum_a e^{f_\theta(s)[a]} - \mathbb{E}_{s \sim \rho_D} \log \sum_a e^{f_\theta(s)[a]} \\
&\quad + \mathbb{E}_{s,a \sim \rho_D} \log \sum_a e^{f_\theta(s)[a]} - \mathbb{E}_{s,a \sim \rho_D} f_\theta(s)[a] \\
&= \mathbb{E}_{s \sim \rho_\theta} \log \sum_a e^{f_\theta(s)[a]} - \mathbb{E}_{s,a \sim \rho_D} f_\theta(s)[a]
\end{aligned} \tag{25}$$

Therefore minimizing $\mathcal{L}_{\text{surr}}(\theta)$ is equivalent to:

$$\text{argmax}_\theta \left( \mathbb{E}_{s \sim \rho_D} \mathbb{E}_{a \sim \pi_D(\cdot|s)} f_\theta(s)[a] - \mathbb{E}_{s \sim \rho_\theta} \log \sum_a e^{f_\theta(s)[a]} \right) \tag{26}$$

From this point onwards, the same strategy for Proposition 2 again applies, completing the proof. $\square$

## B  Experiment Details

**Gym Environments** Environments used for experiments are from OpenAI gym [56]. Table 3 shows environment names and version numbers, dimensions of each observation space, and cardinalities of each action space. Each environment is associated with a true reward function (unknown to all imitation algorithms). In each case, the "expert" demonstrator is obtained using a pre-trained and hyperparameter-optimized agent from the RL Baselines Zoo [61] in Stable OpenAI Baselines [62]; for all environments, demonstration datasets $\mathcal{D}$ are generated using the PPO2 agent [69] trained on the true reward function, with the exception of CartPole, for which we use the DQN agent (which we find performs better than PPO2). Performance of demonstrator and random policies are shown:

| Environments | Observation Space | Action Space | Demonstrator | Random Perf. | Demonstrator Perf. |
|---|---|---|---|---|---|
| `CartPole-v1` | Continuous (4) | Discrete (2) | DQN Agent | $19.12 \pm 1.76$ | $500.00 \pm 0.00$ |
| `Acrobot-v1` | Continuous (6) | Discrete (3) | PPO2 Agent | $-439.92 \pm 13.14$ | $-87.32 \pm 12.02$ |
| `LunarLander-v2` | Continuous (8) | Discrete (4) | PPO2 Agent | $-452.22 \pm 61.24$ | $271.71 \pm 17.88$ |
| `BeamRider-v4` | Cont. ($210 \times 160 \times 3$) | Discrete (9) | PPO2 Agent | $954.84 \pm 214.85$ | $1623.80 \pm 482.27$ |
| `MIMIC-III-2a` | Continuous (56) | Discrete (2) | Human Agent | - | - |
| `MIMIC-III-4a` | Continuous (56) | Discrete (4) | Human Agent | - | - |

Table 3: *Details of Environments*. Demonstrator and random performances are computed using 1,000 episodes.

**Healthcare Environments** `MIMIC-III` is a real-world medical dataset consisting of patients treated in intensive care units from the Medical Information Mart for Intensive Care [63], which records physiological data streams for over 22,000 patients. We extract the records for ICU patients administered with antibiotic treatment and/or mechanical ventilation (5,833 in total). For each patient, we define the observation space to be the 28 most frequently measured patient covariates from the past two days, including vital signs (e.g. temperature, heart rate, blood pressure, oxygen saturation, respiratory rate, etc.) and lab tests (e.g. white blood cell count, glucose levels, etc.), aggregated on a daily basis during their ICU stay. Each patient trajectory has up to 20 time steps. In this environment, the action space consists of the possible treatment choices administered by the doctor every day over the course of the patient's ICU stay, and the "expert" demonstrations are simply the trajectories of states and actions recorded in the dataset. We consider two versions of `MIMIC-III`; one with 2 actions: with ventilator support, or no treatment (`MIMIC-III-2a`), and another with 4 actions: with ventilator support, antibiotics treatment, ventilator support plus antibiotics, or no treatment (`MIMIC-III-4a`).

**Detailed Results** Exact experiment results are shown in Table 4. For each combination of gym environment, imitation algorithm, and dataset size, we follow convention for randomization in our experiment setup by rolling out multiple trajectories ($n_{\text{traj}}$) per trained policy, seeding the experiment multiple times with different expert demonstrations ($n_{\text{demo}}$), and training multiple such policies from different random initializations ($n_{\text{init}}$); see e.g. [12]. Here we set $n_{\text{traj}}=300$, $n_{\text{demo}}=10$, and $n_{\text{init}}=5$. Table 4 shows the means of performance metrics, as well as their standard errors; for ease of comparison, all numbers for gym environments are scaled (according to the performance of demonstrator and random policies given in Table 3) such that the demonstrator attains a return of 1 and the random policy attains a return of 0. For the real-world healthcare environments, we have no access to the ground-truth reward function, and we cannot perform live policy rollouts. We therefore assess imitation performance according to action-matching on held-out test trajectories; see e.g. [64]. In each of $n_{\text{demo}}=10$ folds, we use an 80%-20% train-test split (i.e. 4,666 patients for training, and 1,167 held out for testing). In each instance, we report accuracy of action selection (ACC), area under the receiving operator characteristic curve (AUC), and area under the precision-recall curve (APR).

**Implementations** Wherever possible, policies trained by all imitation algorithms share the same policy network architecture: two hidden (fully connected) layers of 64 units each, followed by ELU activations, or—for Atari—a convolutional neural network with 3 (convolutional) layers of 32-64-64 filters, followed by a fully connected layer with 64 units, with all layers followed by ReLU activations. For all environments, we use the Adam optimizer with batch size 64, 10k iterations, and learning rate 1e-3. Except explicitly standardizing policy networks across imitation algorithms, all comparators are implemented via the original publicly available source code. Where applicable, we use the optimal hyperparameters in the original implementations. The source code for EDM is found at https://bitbucket.org/mvdschaar/mlforhealthlabpub/, and https://github.com/danjarrett/EDM.

**Hyperparameters for EDM** Algorithm 1 is implemented using the source code for joint EBMs [47] publicly available at https://github.com/wgrathwohl/JEM. Instead of Wide-Resnet, for `Acrobot`, `Cartpole`, `LunarLander`, `MIMIC-III-2a`, and `MIMIC-III-4a` we use the fully-connected policy network above, and for `BeamRider` the convolutional neural network above. Specific to EDM are the joint EBM training hyperparameters, which we inherit from [47, 66]: noise coefficient $\sigma=0.01$, buffer size $\kappa=10000$, length $\iota=20$, and reinitialization $\delta=0.05$. We find that these default settings work well with SGLD step size $\alpha=0.01$; for further EBM training-related discussions, we refer to [47, 48].

**Hyperparameters for VDICE** We take the original source code of [43], which is publicly available at https://github.com/google-research/google-research/tree/master/value_dice. In order to adapt the model to work with discrete action spaces, we use a Gumbel-softmax parameterization for the last layer of the actor network. For `Acrobot`, `Cartpole`, `LunarLander`, `MIMIC-III-2a`, and `MIMIC-III-4a` both the actor architecture and the discriminator architecture has two hidden (fully

|  |  | BC | RCAL | DSFN | VDICE | EDM |
|---|---|---|---|---|---|---|
|  | *Demos* | | | *Average Returns* | | |
| Acrobot-v1 | 1 | $0.796 \pm 0.078$ | $0.422 \pm 0.082$ | $0.062 \pm 0.141$ | $0.857 \pm 0.045$ | $\mathbf{0.896 \pm 0.064}$ |
|  | 3 | $0.976 \pm 0.028$ | $0.832 \pm 0.066$ | $0.227 \pm 0.128$ | $0.947 \pm 0.033$ | $\mathbf{0.998 \pm 0.026}$ |
|  | 7 | $0.981 \pm 0.028$ | $0.975 \pm 0.034$ | $0.489 \pm 0.075$ | $0.953 \pm 0.036$ | $\mathbf{0.999 \pm 0.026}$ |
|  | 10 | $0.986 \pm 0.029$ | $0.990 \pm 0.030$ | $0.601 \pm 0.076$ | $0.967 \pm 0.032$ | $\mathbf{0.999 \pm 0.025}$ |
|  | 15 | $0.994 \pm 0.028$ | $0.997 \pm 0.028$ | $0.825 \pm 0.050$ | $0.976 \pm 0.031$ | $\mathbf{1.000 \pm 0.026}$ |
| CartPole-v1 | 1 | $0.321 \pm 0.026$ | $0.233 \pm 0.036$ | $0.317 \pm 0.013$ | $0.324 \pm 0.018$ | $\mathbf{0.428 \pm 0.019}$ |
|  | 3 | $0.607 \pm 0.048$ | $0.586 \pm 0.043$ | $0.373 \pm 0.073$ | $0.738 \pm 0.028$ | $\mathbf{0.900 \pm 0.029}$ |
|  | 7 | $0.819 \pm 0.041$ | $0.894 \pm 0.027$ | $0.523 \pm 0.081$ | $0.867 \pm 0.022$ | $\mathbf{0.982 \pm 0.011}$ |
|  | 10 | $0.932 \pm 0.026$ | $0.991 \pm 0.007$ | $0.458 \pm 0.047$ | $0.967 \pm 0.013$ | $\mathbf{1.000 \pm 0.001}$ |
|  | 15 | $0.997 \pm 0.003$ | $\mathbf{0.998 \pm 0.001}$ | $0.653 \pm 0.074$ | $0.995 \pm 0.004$ | $0.998 \pm 0.002$ |
| LunarLander-v2 | 1 | $0.575 \pm 0.071$ | $0.540 \pm 0.090$ | $0.229 \pm 0.104$ | $0.255 \pm 0.071$ | $\mathbf{0.633 \pm 0.081}$ |
|  | 3 | $0.869 \pm 0.055$ | $0.875 \pm 0.055$ | $0.698 \pm 0.050$ | $0.385 \pm 0.063$ | $\mathbf{0.889 \pm 0.069}$ |
|  | 7 | $0.938 \pm 0.035$ | $0.914 \pm 0.057$ | $0.776 \pm 0.053$ | $0.411 \pm 0.063$ | $\mathbf{0.956 \pm 0.044}$ |
|  | 10 | $0.961 \pm 0.035$ | $0.952 \pm 0.047$ | $0.887 \pm 0.042$ | $0.418 \pm 0.059$ | $\mathbf{0.966 \pm 0.040}$ |
|  | 15 | $0.968 \pm 0.028$ | $\mathbf{0.970 \pm 0.028}$ | $0.913 \pm 0.032$ | $0.417 \pm 0.054$ | $\mathbf{0.970 \pm 0.033}$ |
| BeamRider-v4 | 1 | $0.124 \pm 0.168$ | $0.304 \pm 0.195$ | $0.000 \pm 0.340$ | $0.180 \pm 0.159$ | $\mathbf{0.486 \pm 0.235}$ |
|  | 3 | $0.147 \pm 0.179$ | $0.461 \pm 0.227$ | $0.008 \pm 0.376$ | $0.332 \pm 0.205$ | $\mathbf{0.790 \pm 0.277}$ |
|  | 7 | $0.270 \pm 0.179$ | $0.547 \pm 0.239$ | $0.140 \pm 0.463$ | $0.312 \pm 0.175$ | $\mathbf{0.839 \pm 0.289}$ |
|  | 10 | $0.308 \pm 0.168$ | $0.668 \pm 0.279$ | $0.153 \pm 0.329$ | $0.534 \pm 0.227$ | $\mathbf{0.925 \pm 0.278}$ |
|  | 15 | $0.401 \pm 0.169$ | $0.721 \pm 0.202$ | $0.082 \pm 0.301$ | $0.513 \pm 0.211$ | $\mathbf{0.991 \pm 0.272}$ |
|  | *Metrics* | | | *Action-Matching* | | |
| MIMIC-III-2a | ACC | $0.861 \pm 0.013$ | $0.872 \pm 0.007$ | $0.865 \pm 0.007$ | $0.875 \pm 0.004$ | $\mathbf{0.891 \pm 0.004}$ |
|  | AUC | $0.914 \pm 0.003$ | $0.911 \pm 0.007$ | $0.906 \pm 0.003$ | $0.915 \pm 0.001$ | $\mathbf{0.922 \pm 0.004}$ |
|  | APR | $0.902 \pm 0.005$ | $0.898 \pm 0.006$ | $0.885 \pm 0.001$ | $0.904 \pm 0.002$ | $\mathbf{0.912 \pm 0.005}$ |
| MIMIC-III-4a | ACC | $0.696 \pm 0.006$ | $0.701 \pm 0.007$ | $0.682 \pm 0.005$ | $0.707 \pm 0.005$ | $\mathbf{0.720 \pm 0.007}$ |
|  | AUC | $0.859 \pm 0.003$ | $0.864 \pm 0.003$ | $0.857 \pm 0.002$ | $0.864 \pm 0.002$ | $\mathbf{0.873 \pm 0.002}$ |
|  | APR | $0.659 \pm 0.007$ | $0.667 \pm 0.006$ | $0.665 \pm 0.003$ | $0.673 \pm 0.003$ | $\mathbf{0.681 \pm 0.008}$ |

Table 4: *Detailed Results for Gym and Healthcare Environments.* Bold numbering indicates best performance.

connected) layers of 64 units each with ReLU activation, and—for Atari—the actor and discriminator are replaced with convolutional neural networks with 3 (convolutional) layers of 32-64-64 filters followed by a fully connected layer with 64 units, with all layers followed by ReLU activations. Per the original design, the output is concatenated with the action; this is then passed through 2 additional hidden layers with 64 units each. In addition, to enable strictly batch learning, we set the "replay regularization" coefficient to zero. Furthermore, the actor network is regularized with an "orthogonal regularization" coefficient of 1e-4, actor learning rate of 1e-5, and discriminator learning rate of 1e-3.

**Hyperparameters for DSFN** We take the original source code of [43], which is publicly available at https://github.com/dtak/batch-apprenticeship-learning. Per [37], for Acrobot, Cartpole, LunarLander, MIMIC-III-2a, and MIMIC-III-4a we use a "warm-start" policy network with two shared layers of 128 and 64 dimensions and tanh activation. The hidden layer of size 64 is used as the feature map in the IRL algorithm. Each multitask head in the warm-start policy network has a hidden layer with 128 units and tanh activation. The DQN network (i.e. for learning the optimal policy given a set of reward weights) has 2 hidden (fully-connected) layers with 64 units each, and likewise the DSFN network for estimating feature expectations also has 2 hidden (fully-connected) layers with 64 units. For BeamRider, the first hidden layer in the warm-start policy network is replaced by a convolutional neural network with 3 layers of 32-64-64 filters, and the DQN and DSFN networks are also replaced by the convolutional neural network above. For all environments, the warm-start policy network is trained for 50k steps with the Adam optimizer, learning rate 3e-4, and batch size 64. The DQN network is trained for 30k steps with learning rate 3e-4 and batch size 64 (Adam). Finally, the DSFN network is trained for 50,000 iterations with the learning rate 3e-4 and batch size 32 (Adam).

**Hyperparameters for RCAL** This augments the policy loss with an additional sparsity-based loss on the implied rewards $\hat{R}(s,a) \doteq f_\theta(s)[a] - \gamma \text{softmax}_{a'} f_\theta(s')[a']$ obtained by inverting the Bellman equation [9, 32]. For Acrobot, Cartpole, LunarLander, MIMIC-III-2a, and MIMIC-III-4a we use the fully-connected policy network described above, and for BeamRider the convolutional neural network above. Specific to RCAL is its sparsity-based regularization coefficient, which is set at 1e-2.

**Hyperparameters for BC** The only difference between BC and EDM is the presence of $\mathcal{L}_\rho$, which we remove for our implementation of BC. (Unlike e.g. [32], we do not consider more primitive methods such as linear classifiers/trees to serve as BC, which would not make for a fair comparison/

ablation). For `Acrobot`, `Cartpole`, `LunarLander`, `MIMIC-III-2a`, and `MIMIC-III-4a` we use the fully-connected policy network above, and for `BeamRider` the convolutional neural network above.

**Semi-Supervised Learning** While this is beyond the scope of this work, we briefly note that—by analogy to joint energy-based modeling in general [47]—the EDM algorithm can additionally benefit from semi-supervised learning. Specifically, consider a data-scarce setting where we only have access to limited state-action pairs from the demonstrator—but may have access to additional state-only data. Broadly, this situation arises whenever states are more conveniently observed than actions are. For `CartPole`, Figure 3 shows the results of the original EDM trained on one demonstrator trajectory's worth of state-action pairs, but with access to additional state-only data (**EDM-1t+**) shown in the $x$-axis as multiples of the original amount of state-action data. For comparison, we also reference the performance of EDM without such additional state-only data (EDM-1t), as well as the perfor-

Figure 3: *Semi-Supervised Learning.*

mance of its closest competitor (VDICE-1t), both trained on one trajectory's worth of state-action pairs alone. Notably, observe that (purely by dint of state-only distribution matching) EDM-1t+ manages to extract a sizable gain in performance as the amount of state-only data available increases up to seven-fold. While this improvement is—as expected—less than that conferred by simply adding more state-action trajectories (cf. EDM-3t, which is trained on 3 trajectories' worth of state-action pairs), simply adding state-only data manages to provide as much of a performance boost as the original difference between EDM and VDICE (trained on one trajectory's worth of state-action pairs).

## C  Further Related Work

Throughout this work, we discussed the goal of imitation learning [1–3] in the strictly batch setting, behavioral cloning [4–7] and its relatives [9,30–33], and relationships with the apprenticeship learning family of techniques, including classic (online) inverse reinforcement learning [13–20], (online) adversarial imitation learning [12,22–29], as well as their respective off-policy relatives [34–43,54]. Table 1 summarizes the major aspects of these works as pertinent to our discussion and development.

Further to these works, we also note that another line of research on (online) imitation learning seeks to incentivize the imitating policy to remain within the distribution/support of states encountered in the expert demonstrations [50,55,70–75]. For instance, this is approached through random expert distillation [72], through ensembles of agents [73], or the simple and elegant approach of assigning a unit reward to all demonstrated actions that occur in demonstrated states, and zero otherwise [55]. In general, these methods follow a "two-step" formula, where in the first step some notion of a surrogate reward function is derived/defined, and in the second step this reward function is optimized by way of environment interactions (and as such, they are inherently online techniques). In the same vein, while [75] bears some superficial resemblance to our method by way of energy-based modeling, it is an inherently online technique that depends on training an agent against an explicitly estimated reward function: In the first step, their reward function is defined by modeling the negative energy of the (joint) state-action distribution. However, as with the aforementioned two-step approaches, this must then be followed by an online optimization of this reward function—and is therefore inoperable in our strictly batch setting. Moreover, not unlike in adversarial imitation learning, their KL-divergence minimization interpretation similarly requires the assumption that the optimal reward function is indeed attained—an issue our formulation does not encounter. In contrast, EDM works by decomposing the state-action distribution into an (explicit) policy term and an (implicit) state visitation distribution term, resulting in a single optimization that works in an entirely offline manner.

Finally, tangentially related to our work is a family of inverse reinforcement learning methods designed for reward learning in an offline, model-free setting [76–78]. However, they require access to the demonstrator's policy itself to begin with, and their objective is rather in the inverse problem *per se*—that is, of explicitly recovering the underlying reward function in order to understand behavior.