[Reviews · NeurIPS 2020]

Review 1

Summary and Contributions: This paper proposes a new objective function for batch imitation learning, which can be effectively optimized using only the samples in the batch. This matches three desiderata for strictly-batch imitation learning, proposed by the authors: an explicitly parametrized policy, accounting for rollout dynamics, and operating entirely offline. The algorithm is evaluated on five domains, compared against four competitor algorithms, all of which it outperforms.

Strengths: I think that this setting and problem specification - batch-only imitation, where further interaction with the environment is not allowed - is very compelling, and characterizes many important real-life problems. Specifically focusing on this setting is a major strength. The approach seems reasonable in its details (though I am not an expert in this math), and is thoroughly evaluated.

Weaknesses: I do not see any obvious weaknesses, other than those that apply to all similar RL algorithms.

Correctness: My understanding of the theoretical parts is limited, but they seem fine. I thought the experiment was well done.

Clarity: Yes, but much of the writing is too informal and conversational for a formal publication, or has slightly miscalibrated idioms. "bargains heavily" (presumably "leans heavily on? or relies upon?). "Banks on, "at the end of the day", "so as to speak", "so far so good", are too informal, etc. Parenthetical citations are not nouns.

Relation to Prior Work: Yes

Reproducibility: Yes

Additional Feedback:


Review 2

Summary and Contributions: ============ I have read the authors' feedback. In the reviewers' discussion it seems that a fairly positive consensus among the reviewers was reached, and I hope the paper will be accepted. I encourage the authors to iron out the fine details of the proofs, both in terms of details themselves as well as presentation. ============ The paper proposes a method for performing imitation learning in a pure batch offline setting. It overcomes the difficulties presented by the offline setting by parametrizing the imitator policy using an energy function which allows for computing gradients of its occupancy function through a surrogate loss function. Note on my confidence as a reviewer: I am very comfortable with RL in general, but have had very limited exposure to IRL, so please regard my review as what the paper reads like to a non-IRL expert.

Strengths: The energy based trick the proposed method hinges on is not novel, but based on my understanding from the paper and a literature review I performed, its application to the IRL problem has not been done before. The empirical results presented in the paper are sufficient to demonstrate that it introduce a significant improvement to existing methods in many cases. With some reservations which will be noted later, the paper provides good discussions of key points important for understanding its contribution.

Weaknesses: There are no glaring weaknesses I can spot. I have many suggestions which I think may make the paper stronger or more impactful for the broader RL community, but I think those will be more appropriate in the "additional feedback" section.

Correctness: I did not notice any glaring errors, but I don't have enough expertise in the field to check the paper with the rigor I'd like to.

Clarity: The structure of the paper is solid which contribute a lot to clarity. The wording itself can be awkward at times and I think the writing could use some polishing. In particular, some sentences are written a bit too colloquially, in a way which comes at the expense of clarity.

Relation to Prior Work: Prior work is discussed properly in the sense that a non-expert can easily understand the context. I don't have the expertise to tell if there are any important omissions.

Reproducibility: Yes

Additional Feedback: - The authors note (with references) that the pure behavioral cloning approach performs poorly as it doesn't use information about the dynamics and state distributions of the problem. It would be useful if the authors could present a short concrete example of exactly what type of information is lost when ignoring the MDP structure. - Example for the slightly unclear writing mentioned earlier: in line 107- This sentence is a bit confusing. At a first read it feels like it implies the off-line setting means we have all the information we *need* from the start, which I think is the opposite of what the authors are trying to say. - Line 112 - This sentence immediately brings to mind a decision between parametric vs. non-parametric methods. I don't think that's what the authors are trying to say so maybe the terminology of "parameterizing a policy" should be changed throughout the paper. If it is what the authors are trying to say, then it is not made clear why a parametric approach is the correct choice. - Line 116 - It is unclear to me what "intrinsically batch" means. The sentence seems to imply it's about efficiency, but the discussion at the beginning of page 3 seems to suggest it's more about difficult off-policy evaluation. Which one is it? Furthermore, if it's an offline process, efficiency is usually less important than in an online setting, since there is no need for quick decision making (as long as the process is not too costly as to be computationally prohibitive). - Lemma 1 - The proof is too concise for someone who is not familiar with this exact notation to follow and validate it's accuracy. It seems like the standard technique from statistical mechanics, but different subfields have slightly different notations (especially for defining the partition function), so a more complete version should be placed in the supplementary. - Paragraph starting at line 139 - If the method relies on keeping the computed value of \rho_{\pi_\theta} valid at every iteration, but uses gradients to optimize \theta, wouldn't it be susceptible to integration errors, and therefore the choice of step size. (i.e. accumulate a small error at each gradient step which compounds over time). How do the authors avoid that or know this is not a problem? Empirical results - The authors show convincing results for why their method outperforms previous methods in the batch setting, as that is the setting where their method is designed to perform well. There does not seem to be a problem to apply EDM to the online setting. It would be interesting to see how their method compares with other methods in the online setting, and see if some of the advantages of EDM are still significant in that setting. MIMIC results and real-world domains - For real-world datasets such as the MIMIC III there is often significant stochasticity in the actions selected by clinicians. This raises the issue that accuracy of an imitator policy is not the only important measure, but also how well calibrated a policy is. Can the authors quantify how well calibrated their policies are, as well as remark on their method's ability to imitate stochastic policies. - Shouldn't the discussion start a new section and not be part of section 5?


Review 3

Summary and Contributions: [UPDATE] Thank you for your detailed response. ========== This paper considers a strictly batch imitation learning problem, in which (i) we do not have the knowledge of the transition dynamics, (ii) we do not have access to the environment to do online rollouts during training, and (iii) we only have access to a batch of demonstrations, we cannot further query the expert. First, they have discussed the issues with adapting the classical (online) IRL or adversarial IL algorithms to this off-line setting (mainly due to the standard challenges in off-policy evaluation step). To this end, they propose a novel algorithm called Energy-based Distribution Matching (EDM). They develop this algorithm in a systematic way with proper justifications for the design choices, e.g., minimizing forward KL-divergence between occupancy measures to tackle the fundamental challenges in adapting the online adversarial IL to strictly batch setting. They have also empirically demonstrated the efficacy of their method on: (a) OpenAI gym environments and (b) healthcare application.

Strengths: The paper proposes a novel algorithm to the strictly batch imitation learning problem. The proposed algorithm would be beneficial to domains like healthcare and energy systems, where we have to mainly work with (strictly) off-line data. The paper is very well written, and illustrative experimental results are provided. I have carefully checked the theoretical sections 2-4, and the proposed algorithm is developed in a well-justified manner. Proposition 4 fleshes out a benefit of EDM and a barrier due to strictly batch training. Related works are clearly discussed in the main paper and in Appendix C. The work is well placed in the literature. Reproducibility: Hyperparameter details are clearly reported in the main paper and Appendix.

Weaknesses: I am not fully familiar with the engineering details of the implementation, thus I might have missed some important details in the implementation/experiments. Also, the algorithm in its current form is only applicable to categorical actions, and they have noted the possible extension in the future work section.

Correctness: Yes, I have verfied the theoretical claims.

Clarity: Yes.

Relation to Prior Work: Yes.

Reproducibility: Yes

Additional Feedback:


Review 4

Summary and Contributions: The paper presents an approach for imitation learning in the strict offline setting. By strict offline, they mean a setting where the learner does not have access to the reward function, dynamics of the MDP and no online interaction with the MDP. This severely limits the capabilities of modern imitation learning methods like DAgger (that requires online interaction), inverse reinforcement learning methods (require online interaction and running RL iteratively) and adversarial imitation learning methods (which again require rollouts in the MDP of the learned policy). The only existing baseline (other than adapted variants of online methods) is behavior cloning. The approach proposed by the paper seems to involve approximating the state occupancy distribution of the learned policy using an energy-based model and forcing the state occupancy distributions of the expert and the learned policy to match. The idea of using an energy-based model lets them avoid rolling out the learned policy online to obtain the state occupancy distribution and sample from it directly. They construct a surrogate objective that then uses only the samples from the energy based model and tries to match it to the state occupancy of the expert (which is obtained directly from the dataset.) In addition to state occupancy matching, the other part of the objective is the usual behavior cloning loss which simply maximizes the probability of executing experts action under the learned policy. Hence, the proposed approach tries to imitate the expert policy while ensuring that the state occupancies are close to ensure that the resulting policy has good performance when deployed on the MDP. As far as I am concerned, the core contributions are the idea of using an energy-based model as a proxy for state occupancy matching (couple of issues here that I mention in weakness,) and the construction of the surrogate objective.

Strengths: The strengths of the paper lie in the unique setting that they deal with which can be very beneficial in applications where data is scarce, online interaction during training is expensive/impossible, and it is hard to quantify the reward function/dynamics. I really liked Section 4 as it clearly identified the difference between the proposed method and behavior cloning. It also showed which components of the original objective cannot be accounted for without any online interactions thus showing "kind of a lower bound" on what we can achieve in the purely offline setting. The discussion regarding why simply adapting online imitation learning methods to the strict offline setting would face challenges is also an interesting read.

Weaknesses: The major weakness of the paper is that they oversell/wrongly state the capabilities of the method. I will go into more details in the next few paragraphs. Firstly, I did not understand the proof of Lemma 1. I tried proving it myself by following their sketch, and I just cannot obtain their result. Clearly, the state occupancy distribution is dependent on the MDP dynamics and that is not reflected in the energy-based model approximation they use which is just a function of the policy parameters. As far as I can tell, the Lemma is incorrect in saying that the state occupancy distribution is captured by the energy-based model and the proposed energy-based model formulation is simply an approximation of the true state occupancy distribution of the learned policy. If I am wrong, please help me understand how Lemma 1 proof is correct. Second, it seems proposition 2 is incorrect. I have gone through the proof in the appendix carefully and there is one crucial step (line 504, transition from 3rd equation to 4th equation) where they exchange the gradient and expectation without any explanation, i.e. they say \nabla_\theta E_{\rho_\theta}[f(\theta)] = E_{\rho_\theta}[\nabla_\theta f(\theta)]. This is definitely not true for all distributions and we can easily construct a simple example where it is incorrect (think of a discrete distribution p_\theta over 0, 1 where probability of sampling 0 is \theta, and we are trying to compute the gradient of E_{x ~ p_\theta}[x\theta]). A typical trick to do estimate such quantities involves the log-derivative trick which introduces a log factor, but no such factor appears in their proof. I suggest closely looking at the proof and understanding why it is incorrect. If I am correct, then this means that the statement of the proposition, i.e. gradient of surrogate objective L_\rho is equal to gradient of the log probability under learned policy's state occupancy distribution of states from expert's state occupancy distribution, is incorrect. Again, I am open to any help in understanding how the proof is correct. Coming to the writing of the paper, I think there are several areas where it can be improved. There is a lot of repetition in the claims and arguments all across the paper. Some/most of it can be cut down to make space for background material on energy-based models which I am sure readers (including myself) would greatly appreciate. Without the background material, understanding Algorithm 1 requires a lot of pre-requisite knowledge about energy-based models. Another issue that I have with most recent imitation learning methods is the way they underestimate behavior cloning in empirical comparisons. Behavior cloning works surprisingly well in practice (albeit it has theoretical gaps that are identified in the paper, I think for the benchmarks considered BC should work really well) and the reason why it is not reflected in the paper's experiments is because of the complex policy class chosen. Playing around with simple linear, or kernel policy classes should allow BC to achieve great performance. Instead using a complex policy class consisting of a neural network with 2 hidden layers of 64 units leads to huge overfitting issues for BC and poor performance. I think a better comparison needs to be made and more challenging benchmarks should be chosen (since clearly BC has gaps where it should fail)

Correctness: Have some major issues with claims and proofs in the paper. Look at weaknesses for more details. Have small issues with the empirical methodology of the paper. Look at the last paragraph in weakness for more details.

Clarity: More background material on energy-based models would be greatly appreciated. Please look at the penultimate paragraph in weaknesses for more details. I liked the analysis/interpretation done in Section 4. It is well-written. The arguments as to why naive adaptations of online methods would fail in offline settings are also well-written.

Relation to Prior Work: Yes, there is enough discussion about prior work in terms of broad coverage. The paper could improve on how existing distribution-matching IL approaches relate to the proposed approach. This will make their contributions (other than the new setting) highlighted for the reader to clearly grasp.

Reproducibility: Yes

Additional Feedback: The feedback, comments and questions are dispersed in the previous set of questions in the review. Please look into those carefully. **EDIT AFTER SEEING THE FEEDBACK**: I am convinced that proposition 2 is correct now under the regularity assumptions stated in the feedback. Please include that in the final version and be careful in the rest of the appendix to make sure such assumptions are stated. The feedback does not clearly address my qualms about proof of Lemma 1. It does state that the "decorative and confusing" lemma 1 has been replaced with an introduction that ties with previous work in energy-based models. However, in the light of the fact that Lemma 1 being incorrect, I feel the paper needs to either provide intuition (or analysis) on how good the resulting approximation is. My qualms about using the same architecture for BC are not addressed except saying that "all previous IL works use the same architecture for baselines". However, a practitioner is not restricted to using the same architecture and our work as researchers should be comprehensive enough so that we guide practitioners. Simply conforming to what previous works have done falls short of this standard. I am improving my overall score by 1 point. I hope you take my feedback in a positive light and continue doing such great work in the future.

[Author Response · NeurIPS 2020]

We thank all reviewers for insightful comments. *All existing references are numbered per the bibliography in appendices.*

**[Reviewer 1]** • **Language**: As suggested, we have now standardized the paper for formality and removed colloquialisms.

**[Reviewer 2]** • **Language**: Kindly refer to response for Reviewer 1. • **What information is lost in behavior cloning**:
We have now included (after Line 55) an example pertinent to us: the state visitation distribution of the demonstra-
tor—which results from dynamics—is information in the data, but which BC discounts by focusing only on action
conditionals. • **Line 112, "directly parameterizing a policy"**: You are correct; we are not referring to the parametric/
non-parametric distinction. Instead, we are distinguishing: ① methods that first learn a *reward function* [12–19, 32–37],
thus indirectly inducing a policy (for that reward), vs. ② methods that directly learn a *policy mapping* [11, 20–25, 41].
We have now clarified this (after Line 113). • **Line 116, "intrinsically batch"**: This simply refers to an algorithm that
operates offline *without* recourse to off-policy evaluation (vs. off-policy adaptations of online algorithms). We have
now clarified this (after Line 117), and also explicitly reference Table 1 for its elaboration. • **Energy-based learning**:
You are correct; we rely on a standard technique from statistical physics, and we agree it is more informative—and
responsible—to clearly state this from the get-go. We have now updated the manuscript to properly introduce EBMs
after defining the objective, importantly invoking the joint EBM technique [44–46]—with particular credit to Eq. 7 in
[44] for popularizing the "discriminative + generative" model that is our Eq. 10. This introduction now (appropriately)
replaces the admittedly decorative (and potentially confusing) presentation of Lemma 1. • **Gradient estimator**: Yes,
the choice of step size does affect stability. However, the $\nabla_\theta \hat{\mathcal{L}}_\rho$ update in Algorithm 1 (Line 9) is analogous to that in
standard contrastive divergence, and we inherit any practical implementation details from prior work [44]. • **Online**
**setting**: The online setting is very different: there is no need to approximate state distributions as we do, since they
can just sample from the true distribution directly. We have now included a brief note of this in the Discussion section.
• **Stochastic policies**: Policies per Eq. 6 can readily capture stochasticity. We have now included additional experiment
results in the appendices to measure calibration, which EDM generally *improves* over others: e.g. for `MIMIC-III-2a`,
the expected calibration error is **2.32%** less than BC. • **Discussion section**: Agreed; this is now given its own section.

**[Reviewer 3]** Thank you for your thoughtful comments. For further detail, we will upload code per official guidelines.

**[Reviewer 4]** • **Proof of Proposition 2**: Thank you for pointing out the need for clearer justification for the transition
in Line 504. To be clear, the proposition and algorithm are both correct as intended. We agree, however, that additional
detail would benefit exposition. While you are correct that gradients and expectations cannot be freely exchanged in
the *most general* case, here we can exploit regularity assumptions. To justify the exchange, we have now included a
version of the following (known) result as auxiliary lemma: Let $\theta \in \Theta$, r.v. $s \in \mathcal{S}$, and fix $f: \mathcal{S} \times \Theta \to \mathbb{R}$, where $f(s, \theta)$ is
continuously differentiable w.r.t. $\theta$ and integrable for all $\theta$. Assume for some r.v. $X$ with finite mean that $|\frac{\partial}{\partial \theta} f(s, \theta)| \leq X$
a.s. for all $\theta$. Then $\frac{\partial}{\partial \theta} \mathbb{E}[f(s, \theta)] = \lim_{\delta \to 0} \frac{1}{\delta} (\mathbb{E}[f(s, \theta + \delta)] - \mathbb{E}[f(s, \theta)]) = \lim_{\delta \to 0} \mathbb{E}[\frac{1}{\delta}(f(s, \theta + \delta) - f(s, \theta))] =$
$\lim_{\delta \to 0} \mathbb{E}[\frac{\partial}{\partial \theta} f(s, \tau(\delta))] = \mathbb{E}[\lim_{\delta \to 0} \frac{\partial}{\partial \theta} f(s, \tau(\delta))] = \mathbb{E}[\frac{\partial}{\partial \theta} f(s, \theta)]$, where for equality 3 the mean value theorem guar-
antees existence of $\tau(\delta) \in (\theta, \theta + \delta)$ and equality 4 uses the dominated convergence theorem where $|\frac{\partial}{\partial \theta} f(s, \tau(\delta))| \leq X$
by assumption [Weir, 1973]. Generalizing to gradients simply requires the bound be on $\max_i |\frac{\partial}{\partial \theta_i} f(s, \theta)|$ for elements
$i$ of $\theta$. To be clear, most machine learning models (and energy-based models) meet/assume these regularity conditions or
similar variants; we have also now included a brief note about their reasonableness. • **Relationship with Algorithm 1**:
While Proposition 2 sets the stage for the analysis in Section 4, the (gradient-based) imple-
mentation of Algorithm 1 is *also* correct due to a simpler reason: the batched (empirical loss)
$\nabla_\theta \hat{\mathcal{L}}_\rho$ portion of the update (Line 9) is analogous to that in standard contrastive divergence.

In other words, simply to show it works as intended, we could have stopped at equality 3 in
Line 504 and be done. (Of course, that would have been at the expense of the simplicity of
subsequent derivations for Section 4). • **Energy-based learning**: We agree more background
on EBMs is beneficial. Since we rely on an existing technique in statistical physics, we agree
it is more informative—and responsible—to properly introduce them from the get-go. We
have now updated the manuscript to properly introduce EBMs after defining the objective,
importantly invoking the joint EBM technique [44–46]—with particular credit to Eq. 7 in [44] for popularizing the
"discriminative + generative" model that is our Eq. 10. This introduction now (appropriately) replaces the admittedly
decorative (and potentially confusing) presentation of Lemma 1. Moreover, we have also streamlined the paper by
removing any redundancies/overselling as suggested. • **Approximation of true occupancy**: You are absolutely correct
that the *true* state occupancy distribution depends on MDP dynamics, and that the proposed formulation simply involves
an *approximation* of this distribution. With the updated presentation of joint EBMs (see previous point), this should now
be clear. We now explicitly emphasize it is impossible to obtain the former without actually executing policies online; we
agree the (prior) oversold presentation may be misconstrued to suggest that our offline approach achieves the impossible.
(We also include a note of this contrast/relation to online distribution-matching). • **Environments + comparisons**:
Using common control environments follows recent work in offline IL, and like most (if not all) IL work we use the same
NN architecture for all benchmarks for standard comparison [see e.g. VDICE, DSFN, GAIL]; we also go beyond [32–37]
by adding more complex (`BeamRider`) + realistic (`MIMIC-III`) examples. (Note: MuJoCo is not applicable to categorical
actions). We have also added experiments with Linear BC ("LBC"); see above figure for an example on `LunarLander`.

[Meta-Review · NeurIPS 2020]

All reviewers unanimously agree that the paper makes a nice contribution to imitation learning in the batch setting. That said, the paper has two major weaknesses: 1. The incorrectness of Lemma 1. During the discussion, the reviewers expressed confidence that the authors understand the mistake and know how to address it (see e.g., the post-rebuttal update of R4). Therefore, we are recommending acceptance conditioned on that the authors take this issue seriously, correct the technical mistake, and remove any incorrect or misleading claims associated with it. 2. Learning a model and then doing occupancy matching in the learned model is a simple and likely very strong baseline which the paper does not compare to. The authors are strongly recommended to add such a comparison in the camera-ready version. On a related note, while the algorithm only uses (s,a) pairs as data, trajectory data is often available, from which one can extract (s,a,r,s') pairs. In fact, one of the baselines (VDICE [41]) does require (s,a,r,s') data, and so does the model-based baseline mentioned above. What kind of data is available in practice should be discussed more clearly early in the paper when the problem is set up.